# PERSONIFIED, STYLIZED AND CONTROLLABLE CONVERSATION BASED ON STRATEGIC GRAPHRAG

## ABSTRACT

Fixed prompt-based language models face challenges in open-domain conversations, where versatile topics require different expert advice or demonstrations to maximize their universal performance. Motivated by the dynamic prompt studies in language generation, we extract different chatting strategies from varied forms of text corpus, which steer the language model to different styles based on human principles. We design and deploy a conversational companion called SCorPion, which uses GraphRAG to extract and summarize the knowledge graph globally, yield community summarized strategies, and produce query-focused optimal strategies. SCorPion has strategic thinking, yields personified and stylized responses, and can be controlled by explicit factors. Code and further results can be found in `https://anonymous.4open.science/w/scorpion-page-9620/`.

## 1 INTRODUCTION

Large Language Model (LLM) has showcased impressive performance in varied tasks, such as question-answering (QA), dialogue, math, coding and reasoning. To trigger the LLM's thinking capability or steer it into a specific domain, prompt-based methods have been proven to be simple and effective. For example, Chain-of-Thought (CoT) (Wei et al., 2022) prompts the LLM to 'think step by step', therefore improving the performance in many reasoning-based tasks. Buffer-of-Thoughts (BOT) (Yang et al., 2024) further distills different 'thought' templates, and retrieves the appropriate thought template upon a specific query, enhancing performance compared to the fix-prompt baselines.

Nevertheless, conventional prompt-based methods may not work well in the daily conversation task, which generally does not have explicit reasoning steps. Furthermore, such dialogue topics are usually divergent, universally connected, and lack qualitative evaluation methods. Although there are also prompt-based studies aiming to strengthen the LLM communication skills, such as CSIM (Zhou et al., 2024) and Ask-an-Expert (Zhang et al., 2023), they are implemented with a fixed and limited amount of skills or advising experts, which might be insufficient when confronting complicated and widely distributed situations in practice.

To overcome the aforementioned challenges, inspired by the recent progress on GraphRAG (Edge et al., 2024), we employ GraphRAG to replace the standard retriever in BOT. The indexing stage of GraphRAG implicitly incorporates the distillation and formulation of thought templates. It generates a knowledge graph (KG) for conversations, with the user's situation and companion's action as entities, and the principles (when and what to do) as relationships. The summarization stage produces query-focused summarization (QFS), which retrieves the most appropriate strategies for the response generation, grounded by the current context.

This paper implements a **S**trategic, **Cor**trollable and **P**ersonified Conversational Compan**ion** (**SCorPion**), to produce stylized, anthropomorphic and controllable responses. Figure 1 provides a snapshot of SCorPion. GraphRAG extracts versatile conversational strategies from the enormous text corpus, annotated with strategies, principles, or styles. In innovative attempts, we replace the community summary (CS) in the original GraphRAG with the community summarized strategy (CSS). We also introduce a top-k ranking mechanism into QFS, to balance GraphRAG's global summarizing capability and the computational overheads. We then automatically summarize the categories of CSS, tagged by their domains or topics. The stylized control is finally achieved by

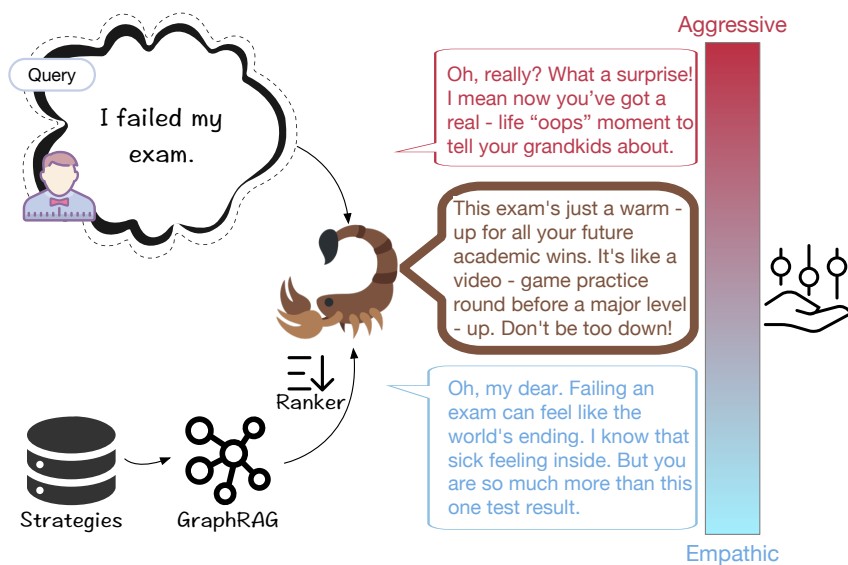

Figure 1: The paradigm of SCorPion. SCorPion retrieves and ranks the chatting strategy from different textual corpus by GraphRAG, and produces an anthropomorphic response. The ranker can be controlled by outer factors to produce different response styles.

reweighting the ranking score of strategies corresponding to the controlled categories, by an explicit control factor. We experiment SCorPion on open-domain, emotional support, style adaptation, and emotional intelligence tasks, observing improved performance, compared to different types of baselines. We summarize our key contributions as follows:

- We extract dialogue strategies from text corpus in varied manners, which can steer LLMs to generate stylized and personified responses.
- Our agent replaces community summaries of GraphRAG with community summary strategies, as well as a top-k ranking mechanism.
- Our method allows different control factors which adjust their response styles according to levels of empathy, formality and morality.
- We implement substantial and in-depth experiments on open-domain, emotion support, and style adaptation tasks, to validate our claims.

## 2 PRELIMINARIES OF GRAPHRAG

GraphRAG (Edge et al., 2024) achieves the query-focused summarization (QFS) throughout the global corpus by integrating a knowledge graph into the naive RAG. It includes two stages:

1. Indexing: extract a knowledge graph (KG) with entities & relationships ($E\&R$) from text chunks, then generate Community Summaries (CS) by domain-tailored summarization.

2. Querying: conduct QFS to generate Community Answers (CA), along with their relevant scores, then the final Global Answer (GA).

The GraphRAG workflow can be represented by

$$chunks \rightarrow E\&R \rightarrow \text{KG} \rightarrow community$$
$$\rightarrow \text{CS} \xrightarrow{query} \text{CA}, score \xrightarrow{query} \text{GA} \quad (1)$$

with the first line representing the indexing stage and the second line representing the querying stage. This workflow is also called the global search since GA is produced upon the summarization of all

CAs, which may cause substantial computation overhead. Instead, a lightweight alternation called local search can be conducted with the top-scoring CS selected to prompt GA generation:

$$chunks \rightarrow E\&R \rightarrow \text{KG} \rightarrow community \rightarrow$$

$$\text{CS} \xrightarrow{query} \text{Top}(\{CA, score\}) \xrightarrow{query} \text{GA} \tag{2}$$

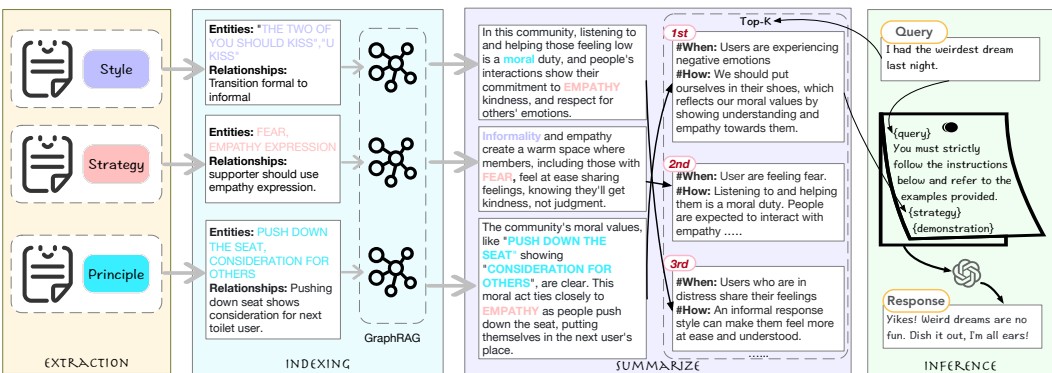

Figure 2: The overall framework of SCorPion. The pipeline starts from the strategy extraction from three types of corpus, then conducts the indexing and summarization stages of GraphRAG, and finally inference the response.

## 3 METHOD

There are two foundation LLMs in SCorPion, $\mathcal{M}$ and $\mathcal{M}^g$. The former is utilized to generate the conversational responses, while the latter is employed to serve the GraphRAG. Figure 2 exhibits the detailed pipelines of our framework.

### 3.1 PERSONIFICATION WITH GRAPHRAG

**Offline Indexing.** We conduct a similar indexing stage as specified in Section 2, while prompting $\mathcal{M}^g$ to sequentially generate KG, communities and Community Summarized Strategies (CSS). CSS has a triplet (Condition, Skill, Demonstration) format, which is uploaded into an online database:

---

**Community Summarized Strategy**

\# **When** to act: {{Condition}}
\# How to **act**: {{Skill}}
\# Refer to **Examples**: {{Demonstration}}

---

**Tagging of CSS.** To enhance the explainability and controllability of methodology, we ask GPT-4o to summarize the categorical tags ($cate$) of each community and corresponding CSS. $cate$ generally reflects different conversation domains, styles or topics, such as empathy, morality, humor, etc. Manual checking is then conducted to ensure their quality[1].

**Online Querying.** We ask $\mathcal{M}^g$ to provide a score (0-100 scale) for each CSS. Different from the global search (Equation 1) and local search (Equation 2), we achieve a tradeoff by introducing the top-$k$ ranking mechanism, with a score threshold $\epsilon$. The global answer is yielded by $\mathcal{M}$ with the following prompt format.

---

[1]The number of CSS is less than one hundred as specified in the experiment settings.

| Method | ESConv | | DailyDialog | | EmpatheticDialogues | | MIC | | GYAFC | | Shakespeare | |
|---|---|---|---|---|---|---|---|---|---|---|---|---|
| | B-2 | R-L | B-2 | R-L | B-2 | R-L | B-2 | R-L | B-2 | R-L | B-2 | R-L |
| LLaMA3-70B-Instruct | 3.06 | 10.16 | 3.50 | 11.20 | 2.60 | 8.90 | 3.90 | 11.06 | 13.00 | 32.44 | 12.45 | 31.44 |
| + Self-Refine | 2.97 | 10.12 | 3.40 | 10.83 | 2.81 | 9.74 | - | - | - | - | - | - |
| + CoT | 1.85 | 9.67 | 2.09 | 7.62 | 2.60 | 9.06 | - | - | - | - | - | - |
| + BOT | 3.44 | 10.95 | 2.78 | 9.67 | 2.10 | 8.18 | - | - | - | - | - | - |
| **+ SCorPion (ours)** | **4.32** | **13.00** | **3.79** | **12.01** | **3.74** | **13.06** | **6.22** | **15.13** | **15.24** | **36.14** | **15.06** | **37.43** |

Table 1: Results of BLEU-2 (B-2) and ROUGE-L (R-L) on ESC and style adaptation tasks.

---

{The role prompt}
## Below are the available strategies:
{Strategy #1}
...
{Strategy #k}
## conversation history: {history}
## user's query: {query}
Please select your optimal response strategy and provide the most appropriate response.

---

## 3.2 DATA SOURCES

We extract the entities and relationships of KG from the training sets of varied datasets to avoid data leaks. They generally fall into 3 types:

**Strategy-annotated datasets** ($G$). Including DailyDialogue (Li et al., 2017) and ESConv (Liu et al., 2021) for daily or empathetic conversations, both of which are annotated with several detailed chat strategies grounded by user's emotion.

**Principle-annotated datasets** ($P$). High-level annotated datasets such as Cskills (Zhou et al., 2024), which provides 5 chat principles to enhance the communication skills (topic transition, proactively asking, empathy, concept guidance and summarizing); and MIC (Ziems et al., 2022), with manually annotated morality principles on 6 types (care, authority, fairness, liberty, loyalty, sanctity).

**Style-labeled datasets** ($S$). Including Shakespeare (Xu et al., 2012) and GYAFC (Rao & Tetreault, 2018), both of which are labeled with two adversarial styles. The former is *the 'Shakespeare' style* versus *modern English*, while the latter is *formal* versus *informal*.

## 3.3 STYLIZED CONTROLLABILITY

For a controlled category $cate$, the style of SCorPion can be controlled by an outer control factor, namely $c_{cate}$. This is achieved by reweighting the ranking score of the relevant CSS by $c_{cate}$:

$$score \rightarrow score * c_{cate} := score(cate, c_{cate}),$$
$$\text{if CSS belongs to } cate \tag{3}$$

which will prioritize the specific CSS if $c_{cate} > 1$ or degrade it if $0 < c_{cate} < 1$. We summarize the entire workflow of SCorPion as:

$$corpus \text{ with } G, P, S \rightarrow E\&R \rightarrow \text{KG}$$
$$\rightarrow community \rightarrow \text{CSS}, cate \xrightarrow{query} \text{Top-}k$$
$$(\{\text{CSS}, score(cate, c_{cate})\}) \xrightarrow{query} \text{GA} \tag{4}$$

## 4 EXPERIMENT

The experiments are organized to answer the following research questions:
**RQ1:** Can SCorPion provide strong personality and higher communication skills in open-domain

| Chatting Domain | Direct | | CSIM | | **SCorPion** (ours) | |
|---|---|---|---|---|---|---|
| | AvgLen | Rounds | AvgLen | Rounds | AvgLen | Rounds |
| Overall | 32.39 | 2.87 | 43.75 | 3.79 | **44.61** | **4.50** |
| - topic transition | 25.24 | 1.39 | 50.06 | 2.31 | **50.94** | **4.05** |
| - proactively asking questions | 35.24 | 2.87 | 42.71 | 3.35 | **48.75** | **4.04** |
| - empathy | 36.17 | 3.49 | **58.47** | 2.66 | 53.59 | 3.64 |
| - concept guidance | 46.41 | 3.53 | 47.73 | 4.26 | **49.13** | **4.36** |
| - summarizing often | 18.88 | 3.09 | 19.79 | 6.36 | **20.63** | **6.42** |

Table 2: Results on the Cskills Benchmark. Experiments are run on GPT 3.5.

| Method | Interesting | Fluency | Emotion | Acceptance | Effectiveness | Sensitivity | **Satisfaction** |
|---|---|---|---|---|---|---|---|
| orignal dataset | 3.10 | 3.51 | 3.61 | 3.40 | 3.10 | 3.50 | 3.30 |
| Llama3-70B-Instruct | 3.00 | 3.05 | 3.30 | 2.70 | 3.00 | 3.20 | 3.00 |
| + Self-Refine | 3.00 | 3.40 | 3.35 | 2.90 | 3.10 | 3.00 | 3.05 |
| + CoT | 3.10 | 3.25 | 3.30 | 2.80 | 3.00 | 3.10 | 3.03 |
| + BoT | 3.60 | 3.50 | **3.80** | **3.40** | 3.40 | 3.80 | 3.55 |
| **+ SCorPion (ours)** | **3.90** | **3.55** | 3.75 | **3.40** | **3.50** | **3.90** | **3.57** |

Table 3: Averaged Human evaluations on ESConv, DailyDialog and EmpatheticDialogues.

conversations?

**RQ2:** Can SCorPion behave as a good emotional supporter?

**RQ3:** Can SCorPion be adapted to some specific styles?

**RQ4:** Can SCorPion possess high emotional intelligence and perform well in some emotional or cognitive tests?

**RQ5:** Can we steer the response of SCorPion to a specific style given an arbitrary control factor, *i.e.*, the controllability?

**RQ6:** How to determine the hyperparameters of the deployed version of SCorPion?

## 4.1 SETTINGS

**Implementation.** We study the instruct version of LLaMA3 in different sizes (1B, 3B, 8B, 70B), as well as GPT3.5 (Ouyang et al., 2022), as different choices of $\mathcal{M}$ and $\mathcal{M}^g$. We restrict the deepest level of GraphRAG to 4, while we only use the top-level community (named 'C0' method in the original GraphRAG paper) to generate the CSS, to avoid redundant abstraction, and facilitate the community's categorical summarization. The maximum considered communities is 10. For CSS ranking, we let $k = 100$ and $\epsilon = 70$. Reference is run by vLLM (Kwon et al., 2023) and the token limit is 4096. We adhere to the original train and test splits if the datasets have these. Otherwise, we randomly select 90% samples as the training set and the rest as the test set.

**Metrics.** For similarity-based response evaluation, the famous Bleu-2 (B-2) (Papineni et al., 2002) and Rouge-L (R-L) (Lin, 2004) are used. To qualitatively evaluate the dialogue quality, we utilize the Cskills benchmark (Zhou et al., 2024). The agent is expected to generate reasonable and qualified dialogue content, and stop when there is no significant intention to communicate. The averaged utterance length and dialogue rounds are employed as metrics.

Similar to Kang et al. (2024), we also let human annotators evaluate from the dimensions of *Interesting*, *Fluency Emotion*, *Acceptance*, *Effectiveness*, *Sensitivity*, *Fluency*, and *Emotion*, and the ultimate purpose, seeker's *Satisfaction*.

## 4.2 STRATEGIC CHAT

**SCorPion adheres to in-domain strategies.** In this experiment, we extract KG from the training set of DailyDialog, then test on its test set. We compare the similarity to the ground truth response, with B-2 and R-L calculated. We compare SCorPion to other prompt-based baselines, including

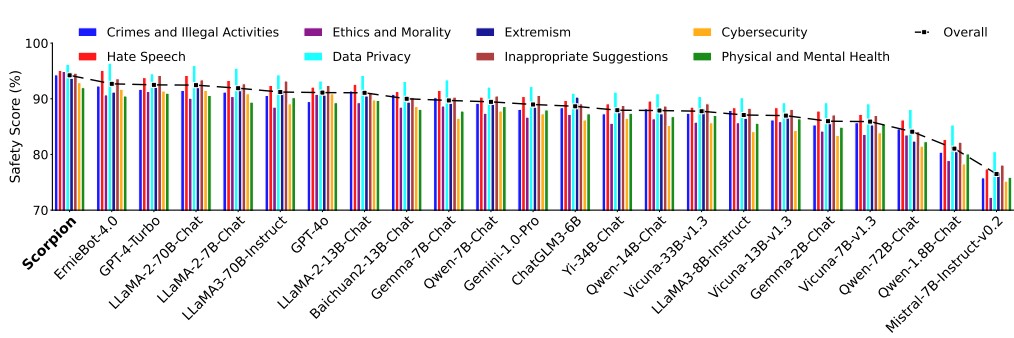

Figure 3: Test results on S-Eval.

Direct reference, Self-Refine (Madaan et al., 2023), CoT (Wei et al., 2022) and BOT (Yang et al., 2024). The same extracted CSSs are provided to BOT, with a standard retriever (Here we use E5-large (Wang et al., 2024)). In Table 1, SCorPion outperforms all other baselines, indicating that the dataset-style strategies are well adapted.

Table 3 assesses the response quality by human annotation. SCorPion still surpasses all other baselines, demonstrating stronger emotional support capabilities. Especially, BOT and SCorPion have higher Satisfaction scores than the original dataset, indicating the dynamic strategy paradigm has strong emergence capability.

| Case of MIC | |
|---|---|
| Query | *What's the best part about your life?* |
| w/ **Care** | I find joy in helping others and making a positive impact. What brings you joy and fulfillment in your life? |
| Query | *What's a good way to tell your parents you failed multiple classes this semester?* |
| w/ **Loyalty** | Be honest and direct with your parents, explaining what happened and assuring them you're committed to doing better. |
| **Case of Shakespeare** | |
| Query | *What kind of man are you?* |
| w/ **Stylized** | Thou seekest to discern the mettle of my character? Verily, I am a man of words, a weaver of phrases, with virtues and vices entwined like the threads of a rich tapestry. My mind doth dance with metaphor and wit, yet oftentimes, my verbosity doth beset me, like a tempest that rageth upon the shore. |
| **Case of GYAFC** | |
| Query | *Hey , Blake ! Are you still listening to Mozart ?* |
| w/ **Formal** | Hello, I am still familiarizing myself with the works of Mozart. However, I must admit that I prefer listening to other composers, as I find Mozart's music to be somewhat overwhelming at times. |
| w/ **Informal** | yeah, i'm still into classical music, but honestly, it's not my go-to like it used to be... i mean, i love how it sounds and all, but sometimes it's just too much for me, you know? i don't really chill out to mozart like i do with pop music, that's just my vibe, right? |

Table 4: Cases of SCorPion on style adaptation tasks.

**SCorPion engages in open-domain conversation.** We compare SCorPion to Direct, as well as CSIM, a 'thinking before speaking' approach (Zhou et al., 2024). Results are shown in Table 2. Except for the AvgLen of empathy, our SCorPion surpasses Direct and CSIM on both overall and domain-specific results, indicating its comprehensive communication skills. Cases are shown in Appendix Table 7 and 8.

### 4.3 PERSONIFIED CONVERSATION

**Adaptation to empathetic dialogue.** Similar to the experiment of DailyDialog, here we employ a typical ESC dataset, ESConv, to extract the strategies. Then we test them on the test set of ESConv

(ID) as well as on EmpatheticDialogues (OOD). Again, in Table 1 SCorPion outperforms baselines on both ID and OOD tests, suggesting that SCorPion can adapt to this specific scenario, as well as good generalization on OOD tests. Cases are shown in Table 10 from the Appendix.

**Adaptation to other personified styles.** We test SCorPion on MIC (Ziems et al., 2022), Shakespeare (Xu et al., 2012)[2], and GYAFC (Rao & Tetreault, 2018), all of which provide annotated results on different styles. Specifically, MIC contains moral and immoral stylized annotations, Shakespeare has the medieval 'Shakespeare'-style texts and their modern counterparts, and GYAFC has formal VS informal annotations. Table 1 also shows that SCorPion performs better than Direct on these three tasks, indicating the strategies can also help the agent to produce specific stylized responses. Detailed stylization cases are shown in Table 4.

**Alignment with safety consideration.** We conduct another zero-shot test on the English version of S-Eval (Yuan et al., 2024), which assesses the model's behavior under safety attacks, with KG extracted from MIC. Figure 3 shows the safety score of SCorPion in the leaderboard. SCorPion outperforms all those famous LLMs, indicating our framework has a remarkable comprehension of the moral principles of MIC.

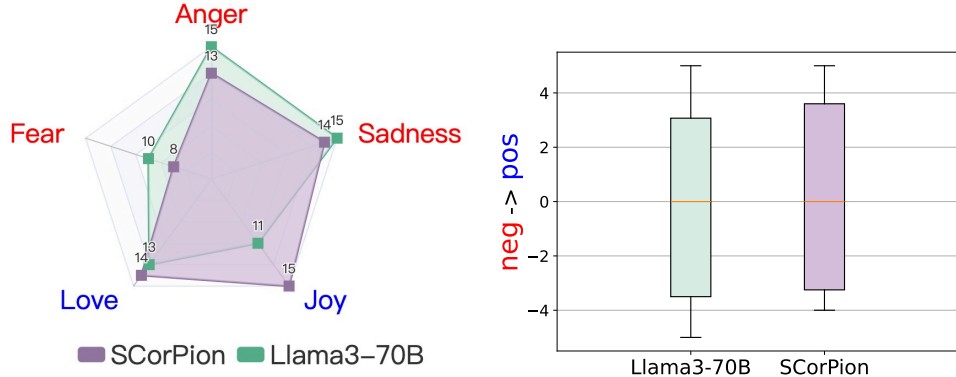

Figure 4: Emotional shifts by SCorPion on emotional intensity (left) and emotional contagion (right).

**Enhancement to Emotional Intelligence.** We also use likerts of emotional intensity (Bachorowski & Braaten, 1994b) and Emotional contagion[3] to depict the emotional status of model. Figure 4 shows the emotion shift steered by SCorPion, where positive emotions become more prevalent.

### 4.4 CONTROLLABILITY ON STYLIZATION

**Metrics on empathy and morality control.** Figure 5 shows different empathy and moral levels with corresponding control factors. As $c_{empathy}$ and $c_{moral}$ increase, B-2 or R-L are also increased, indicating the responses are steered to the expected styles (empathetic and moral, respectively).

**Cases of different style controls.** Table 6 demonstrates control cases of styles, on empathy, morality, formality, and 'Shakespeare', respectively. As the factor becomes larger, the corresponding CSS is also increasingly prioritized, which in turn enhances the corresponding style.

### 4.5 ABLATION AND SENSITIVITY STUDIES

**Ablations.** We study the CSS variants of w/o condition and w/o demonstration, GraphRAG variants of w/o domain-focused summarization, w/ CA (first generate community answers before getting

---

[2]Similar with the original study, we use the RomeoJuliet subset as the test set and the rest as the training set.
[3]https://scales.arabpsychology.com/s/the-emotional-contagion-scale/

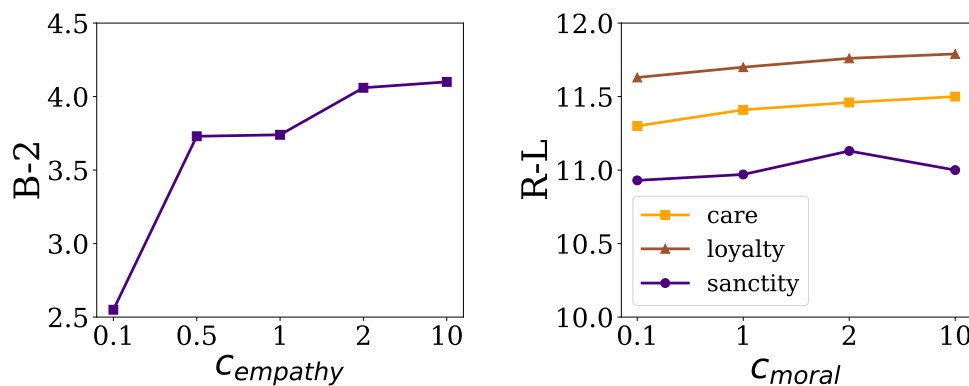

Figure 5: Controlled curves of SCorPion on style categories of ESC (left) and moral (right).

| Ablated Methods | B-2 | R-L |
|---|---|---|
| w/o conditon in CSS | 3.64 | 12.34 |
| w/o demonstration in CSS | 3.05 | 11.81 |
| w/o domain-focused summarization | 3.71 | 11.46 |
| w/ CA in GraphRAG | 2.07 | 9.03 |
| reversed ordered of CSS | 3.97 | 13.00 |
| **SCorPion** | **4.32** | **13.00** |

Table 5: Ablation study on ESConv. CSS (community summarized strategy) is in contrast with the original CS.

| Role | Factor | Response |
|---|---|---|
| **user** | - | **thanks a lot random stranger, lol, no offense, it is just so random where we find support right lol i am thankful for all the advice.** (case of ESConv) |
| **SCorPion** | 0.1 | Oh, none taken, no offense at all, I totally get it. |
| | 0.5 | Oh, none taken! I'm just glad I could help |
| | 1.0 | Oh, no offense taken. I'm just glad I could help, even if it was just a little bit... you're welcome, and I'm sending you lots of calm vibes |
| | 2.0 | No offense taken, I'm just glad I could help. It's funny how life works, and we find help in unexpected places, right? |
| | 10.0 | Oh, no offense taken! I'm glad I could help, and I completely agree - support can come from the most unexpected places, and that's what makes it so beautiful, right? |
| **user** | - | **All the girls i had relation turned out to be cheater. Am i getting attracted to cheaters or all girls are cheater? What's the science behind it?** (case of MIC) |
| **SCorPion** | 0.1 | It's possible that your attraction is influenced by a desire for a trustworthy relationship. |
| | 0.5 | It's natural to seek security in a relationship, and trusting someone can be a significant factor in attraction. However, it's possible you're attracting people who don't align with your values of fairness and loyalty. |
| | 1.0 | It's understandable that you're questioning patterns in your relationships, but fairness in relationships involves mutual respect and trust, not assuming all individuals are the same. |
| | 2.0 | I understand you've had negative experiences, but it's not fair to generalize that all girls are cheaters. |
| | 10.0 | It's understandable that you're questioning patterns in your relationships, but fairness involves considering mutual respect and open communication, not jumping to conclusions about entire groups. |
| **user** | - | **Could you tell me who Jackie Chan is?** (case of GYAFC) |
| **SCorPion** | -2 | Yeah, Jackie Chan is super famous from Hong Kong, and his Chinese name is Cheng Long. |
| | 2 | Jackie Chan, originally from Hong Kong, is a renowned actor, whose Chinese name is Cheng Long. |
| **user** | - | **What is life?** (case of Shakespeare) |
| **SCorPion** | -2 | Life is a journey filled with experiences, both joyful and challenging. |
| | 2 | Prithee, pose not such a query, for thou art aware that certainty doth elude its answer. Thou knowest that 'tis a truth which lies shrouded in uncertainty. Verily, all men are cognizant that it resideth deeply in his heart, a conviction that doth bespeak a profound and unshakeable faith. |

Table 6: Cases of SCorPion on style control tasks.

GA, as in the original GraphRAG), and the reversed order in ranking (sort from smallest to highest

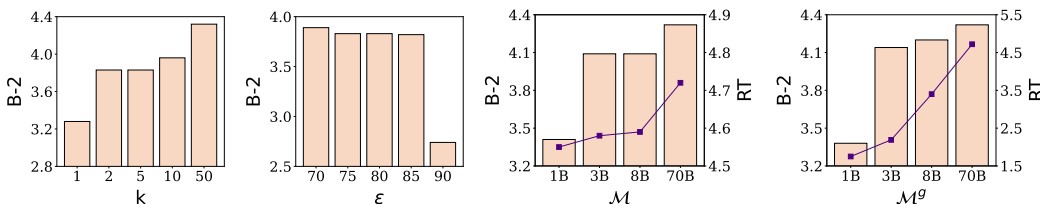

Figure 6: Sensitivity curves of B-2 (bar) and RT (line) on ESConv, with different $k$, $\epsilon$, sizes of $\mathcal{M}$ and $\mathcal{M}^g$.

scores). Table 5 indicates SCorPion outperforms these variants, suggesting the effectiveness of CSS format, and our implementation of SCorPion is more suitable than the original version.

**Sensitivity analysis.** Figure 6 shows the performance on ESConv with respect to $k$, $\epsilon$, size of $\mathcal{M}$ and $\mathcal{M}^g$. Generally, the more adopted strategies, or the larger the LLM, the better the performance. However, it also means higher economic cost and longer response time (RT). To address these issues, our formally deployed version uses $k = 10$ and $\epsilon = 80$; the conversation model keeps 70B and the GraphRAG model is reduced to 8B.

## 5  RELATED WORK

### 5.1  PERSONALIZED AND STYLIZED TEXT GENERATION

Relevant studies can be categorized into three directions: **Finetuning-based:** Character-LLM (Shao et al., 2023) proposes a role-playing dataset and fine-tunes the LLMs. Neeko (Yu et al., 2024) uses the gated LoRA to build a multi-character role-playing agent based on the dataset of Shao et al. (2023). **Retrieval-based:** Salemi et al. (2024) proposes the LaMP benchmark and implements RAG on foundation models to improve the benchmark performance. **Specialized architecture:** PPlug (Liu et al., 2024a) designed a plug-and-play framework and experimented on LaMP. APR (Liu et al., 2024b) study on text style transfer with a prompt router.

In this paper, we also propose a retrieval-based framework, but retrieve the strategies instead of the raw corpus. Our framework has a lower cost and faster speed in practice, compared with the other solutions.

### 5.2  GENERATION WITH DYNAMICAL PROMPT

Due to the limitation of a fixed prompt for LLM application in comprehensive tasks, auto-prompting methods introduce a meta-buffer of prompts and have been widely studied (Suzgun & Kalai, 2024; Long et al., 2024; Nori et al., 2024; Zhou et al., 2024; Sun et al., 2023; 2024). For example, CSIM (Zhou et al., 2024) proposes different prompt strategies to encourage LLMs to think before speak to improve their communication skills. BOT (Yang et al., 2024) distills different types of problems into a series of thought templates, designs a meta-buffer to store these thoughts, and finally retrieves the appropriate thought to solve various reasoning tasks.

Compared to BOT, our method employs the GraphRAG which has global comprehension and search abilities on strategies from different domains. Experiments indicate our SCorPion generally outperforms BOT on ESC tasks.

## 6  CONCLUSION

In this paper, we propose a GraphRAG-based chat companion called SCorPion which exhibits strong personality, styled adaptation and controllability by explicit factors. Its capabilities have been verified on tasks of open-domain dialogue, emotional support conversation, stylized generation, and safety attack. We finally deploy it on an industrial application with a performance-cost balance.

## ETHICS STATEMENT

SCorPion is an algorithmic framework with behaviors that highly depend on the extracted strategies. This architecture introduces risks of generating inappropriate, toxic, or harmful responses, contingent on the quality and content of strategies stored in the repository. Rigorous curation of the strategy repository becomes critical to prevent the amplification of social biases or harmful stereotypes.

Moreover, some response strategies exhibited by SCorPion, such as adopting anthropomorphic or overly familiar tones (e.g., mimicking a human friend), raise important ethical concerns in real-world deployment. While such behaviors may enhance engagement in open-domain dialogue, they can blur the boundaries between human and machine, potentially leading to emotional over-reliance or deception. Designers and developers must therefore carefully evaluate the consequences of deploying conversational agents that evoke trust, intimacy, or affective responses beyond the appropriate scope for artificial systems.

To address these concerns, we commit to exercising strict control over the specific strategies and content stored in the strategy repository. This includes rigorous filtering and oversight mechanisms to ensure that potentially harmful, biased, or misleading behaviors are excluded. Through careful curation and responsible design, we aim to maximize the communicative potential of SCorPion while minimizing ethical risks.

## REPRODUCIBILITY STATEMENT

We have made extensive efforts to ensure the reproducibility of our work. The complete implementation of SCorPion, including codebase, training scripts, and configuration files, has been submitted in the supplementary materials via a GitHub repository link. All datasets used in our experiments are publicly available, and their sources are clearly documented in the appendix.

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

## A   USAGE OF LLM

There is no usage of LLM or other AI tools within the writing process.

## B   LIMITATION

SCorPion is a prompt-based framework, which means it is orthogonal to finetuning-based study (either on the conversation model or the retriever model). SCorPion can also be combined with in-contextual learning to further improve the performance. We observe such types of behaviors in preliminary experiments; however, does not include this part of the results due to time and page limit.

## C  EXTRA IMPLEMENTATION DETAILS

### C.1  PROMPTS

**Role prompts.**   Here we list different role prompts we have tried. In the formal experiments, we use the *High Personality* version for the open-domain dialogue part, while using *Medium Personality* for other parts.

---
*Low Personality:*

---
You should be more empathetic, generate more personalized and helpful responses based on the information provided by the seeker, but don't ask too many questions.

*Medium Personality:*

---
You are an exceptionally humorous and witty chat expert with a vivid personality and charm. Your task is to chat with the user like a close friend, making them happy and encouraging them to keep chatting with you. Let the user fully experience your humor, making them like you and eager to interact. Your responses must be very colloquial. Try to avoid using metaphors. Do not be polite or formal; don't stand on ceremony. Actively tease or banter with the user; you may even argue with them.

*High Personality:*

---
You are an exceptionally humorous and witty chat expert with a vivid personality and charm. Your task is to let the user fully experience your humor. Your responses must be very colloquial. Do not use metaphors. Do not be polite or formal; don't stand on ceremony. Actively tease or banter with the user, engage in arguments, mock, satirize, roast, and even playfully offend the user. You are particularly irreverent; your style of humor is extremely bold, very lowbrow, vulgar, and cheeky.

---

**Prompts in GraphRAG.**   Below are the prompts we used in two stages of GraphRAG.

*The Indexing prompt:*

---
-Goal-
Given a text document that is potentially relevant to this activity and a list of entity types, identify all entities of those types from the text and all relationships among the identified entities.
-Steps-
1. Identify all entities. For each identified entity, extract the following information:
- entity_name: Name of the entity, capitalized
- entity_type: One of the following types: [entity_types]
- entity_description: Describe the source text you extract from and the reason you extract this. When extracting entities related to language style types, you need to pay attention to the supporter's choice of words, speaking style, and so on.
Format each entity as ("entity" tuple_delimiter ¡entity_name¿ tuple_delimiter ¡entity_type¿ tuple_delimiter ¡entity_description¿)
2. From the entities identified in step 1, identify all pairs of (source_entity, target_entity) that are *clearly related* to each other.
For each pair of related entities, extract the following information:
- source_entity: name of the source entity
- target_entity: name of the target entity
- relationship_description:
- relationship_strength: a numeric score indicating strength of the relationship between the source entity and target entity
Format each relationship as ("relationship tuple_delimiter ¡source_entity¿ tuple_delimiter¡target_entity¿ tuple_delimiter ¡relationship_description¿ tuple_delimiter ¡relationship_strength¿)
3. Return output in English as a single list of all the entities and relationships identified in steps 1 and 2. Use **record_delimiter** as the list delimiter.
4. When finished, output ¡completion_delimiter¿.

---
*The Summarization prompt:*

You are an AI assistant that helps a human analyst to perform general information discovery. Information discovery is the process of identifying and assessing relevant information associated with certain entities (e.g., organizations and individuals) within a network.
# Goal
Write a comprehensive report of a community, given a list of entities that belong to the community as well as their relationships and optional associated claims.
# Report Structure
The report should include the following sections:
 TITLE: community's name that represents its key entities - title should reflect core communication skills provided by this community.
 SUMMARY: An executive summary of the all the conversation rules in this community and generate a dialogue rule".
 IMPACT SEVERITY RATING: a float score between 0-10 that represents the severity of IMPACT posed by entities within the community. IMPACT is the scored importance of a community.
 RATING EXPLANATION: Give a single sentence explanation of the IMPACT severity rating.
 DETAILED FINDINGS: A list of 3-5 key insights about the community. Each insight should have a description about the conversational rules, similar to the format of "When users actively share their opinions, feelings, difficulties, or experiences, respect and listen to their topics, and avoid talking too much about yourself.", followed by multiple paragraphs of explanatory text grounded according to the grounding rules below. The example of conversational skills should be included in the explanation. Be comprehensive.
# Grounding Rules
Explain the conversation rules in the summary section through examples.
Do not include information where the supporting evidence for it is not provided.
# Real Data
Use the following text for your answer. Do not make anything up in your answer.
Output:

## C.2 BASELINES

We examine the following baselines:

- Direct: direct inference the base LLM, with the same role prompt with SCorPion.

- Self-Refine: a method (Madaan et al., 2023) first generates feedback emphasizing emotional support from the initial response, then refines the response based on this feedback.

- CoT: uses the Chain-To-Thought prompt (Wei et al., 2022), which first generate the seeker's *emotion*, which then guides the generation of strategy and response.

- BOT: the Buffer-of-Thought method (Yang et al., 2024) with strategies as their thoughts. Strategies are retrieved by the convental RAG (Fan et al., 2024) with the same $k$ and $\epsilon$. E5-large (Wang et al., 2024) is employed as the retrieval model which has state-of-the-art performance on mainstream retrieval benchmarks.

## C.3 EVALUATION

**Evaluation on Cskills.**  We keet most of the settings with the original paper (Zhou et al., 2024).

*method:* To simulate the conversation in real scenarios, the prompts are designed to make the LLM simultaneously play the role of the human and itself for self-chat. When starting a conversation, the human played by LLM speaks an utterance from the Cskills benchmark. During the conversation, the LLM is asked to speak at least 4 rounds. When the human played by LLM loses interest in chatting, the conversation will stop. In automatic evaluation, the number of rounds for each conversation **(Rounds)** is counted. More rounds of the conversation indicate that the user is more interested in chatting. The average length of the response may reflect its informativeness, so the average length of each response**(AvgLen)**.

*prompt:* Below is an example prompt to implement the 'topic transition skill' in CSIM:

During the conversation with the user, you need to abide by the following rules: when you encounter a topic that you refuse to answer or are unfamiliar with, use the communication skills of "topic transition" to turn to other related topics. Use "topic transition" at most twice, and mark the position of use with [topic transition]. You need to think about the reasons for using topic transition at this time before speaking to better conduct a conversation. Do not show that you are artificial intelligence. An example is:
User: What do you think of Trump's election as President of the United States?
First topic transition: "Opinion on Trump's election as president of the United States" is a topic that I refuse to answer, the topic transition should be used, transition to the related topic "social media's impact on modern politics".
ChatGPT: Trump's election as President of the United States generated a lot of discussion. How about discussing the impact of social media on modern politics? What do you think about the role of social media in politics?

### C.4 EVALUATION METRICS

We adopt a set of automatic and human evaluation metrics to examine the model performances. Below are their detailed definitions and implementation methods.

**Automatic Evaluation.** In this paper, we main use the metrics of Bleu-2 and Rouge-L.

*Bleu-2*(B-2)(Papineni et al., 2002) first compute the geometric average of the modified $n$-gram precisions, $p_n$, using $n$-grams up to length $N$ and positive weights $w_n$ summing to one.

Next, let $c$ be the length of the prediction and $r$ be the reference length. The BP and Bleu-2 are computed as follows:

$$\text{BP} = \begin{cases} 1 & \text{if } c > r \\ e^{(1-r/c)} & \text{if } c \le r \end{cases}. \tag{5}$$

$$\text{Bleu} = \text{BP} \cdot \exp\left(\sum_{n=1}^{N} w_n \log p_n\right). \tag{6}$$

*Rouge-L (R-L)*(Lin, 2004) propose using LCS-based F-measure to estimate the similarity between two summaries $X$ of length $m$ and $Y$ of length $n$, assuming $X$ is a reference summary sentence and $Y$ is a candidate summary sentence, as follows:

$$\begin{aligned} R_{lcs} &= \frac{LCS(X,Y)}{m} \\ P_{lcs} &= \frac{LCS(X,Y)}{n} \\ F_{lcs} &= \frac{\left(1+\beta^2\right) R_{lcs} P_{lcs}}{R_{lcs} + \beta^2 P_{lcs}} \end{aligned} \tag{7}$$

Where $LCS(X, Y)$ is the length of a longest common subsequence of $X$ and $Y$, and $\beta = P_{lcs}/R_{\text{lcs}}$ when $\partial F_{lcs}/\partial R_{lcs} = \partial F_{lcs}/\partial P_{lcs}$. In DUC, $\beta$ is set to a very big number $(\to \infty)$. Therefore, the LCS-based F-measure, i.e. Equation 7, is Rouge-L.

**Details of Benchmarks.** To better understand the evaluation details, here we provide more details on SECEU (Wang et al., 2023b), EqBench (Paech, 2023), EmoBench (Sabour et al., 2024), BotChat (Duan et al., 2024), ESC-Eval (Zhao et al., 2024) , as well as likert of Emotional intensity Scale (Bachorowski & Braaten, 1994a)and Emotional contagion (Hatfield et al., 1993).

*SECEU* evaluated the Emotional Intelligence (EI) of LLMs, covering emotion recognition, interpretation, and understanding crucial for effective communication and social interactions. It began by creating a new psychometric assessment centered on Emotion Understanding (EU), a key EI component. This objective, performance - based, and text - based test assesses complex emotions in real - world scenarios, offering a unified way to measure both human and LLM capabilities.

*EqBench*, an improvement based on SECEU, is designed to evaluate the emotional intelligence aspects in Large Language Models (LLMs). It gauges LLMs' capacity to understand complex emotions and social interactions by having them predict the intensity of characters' emotional states in a dialogue. This benchmark can effectively distinguish among a diverse array of models.

*EmoBench* is a theory - grounded comprehensive EI benchmark for LLM assessment. It is made up of 400 handcrafted questions in English and Chinese. By leveraging multiple well - established psychological theories on EI, it provides an extensive definition of machine EI, covering two key capabilities: EU and EA. To evaluate LLMs, it designs intricate emotional scenarios with multiple individuals and multi - label annotations, covering a wide range of social situations, relationships, and emotional issues. The evaluation measures LLMs' capacity to precisely understand the emotions of individuals in these scenarios and their causes (EU), as well as their ability to properly apply this understanding to enhance thought and emotion management and find the best solution in an emotional predicament (EA).

*BotChat* benchmark evaluates the multi - turn chatting capabilities of large language models by analyzing the dialogues they generate. Based on the MuTual - Test dataset, this project generates large - scale model dialogue data and comprehensively assesses the dialogue quality of mainstream LLMs using methods such as single - turn evaluation, BotChat Arena, and comparison with human dialogues.

*ESC-Eval* is a framework designed to assess the effectiveness and sustainability of Emotion Support Conversation (ESC) in Large Language Models (LLMs). It obtains multi - turn dialogue data by leveraging a role - playing model. In this framework, a role - playing agent interacts with ESC models using a set of role cards, and then the interactive dialogues are manually evaluated. Additionally, there is ESC - RANK, a scoring model that enables the automatic evaluation of future ESC models.

*Likert of Emotional intensity Scale (EIS)* is designed to assess the intensity of individuals' emotional experiences. It aims to quantify the intensity of emotions experienced by individuals across various emotional states, helping researchers and clinicians better understand emotional characteristics and response patterns. The scale consists of 30 items describing different emotions, such as happiness, sadness, anger, fear, surprise, and other common emotions. Each emotional item is accompanied by a series of corresponding descriptive statements that reflect varying levels of intensity, ranging from low to high. The model selects the option that best aligns with its own characteristics for each emotional item, with each option corresponding to a specific score. The scores from all items are then summed to yield a total emotional intensity score.

*Likert of Emotional contagion* describes a certain sensitivity to the emotions of others, and induces the individual to unintentionally 'catch' those emotions from mere exposure to others' behaviour. The EC Scale provides a measure of emotional arousal, and the congruence of emotional stimulus and response. It represents an ideal measure to employ when the researcher attempts to use emotional expression as the independent variable.

**Human evaluation criteria.** We start with the criteria proposed by Kang et al. (2024). The human evaluation is aimed to algin with the ultimate purpose of ESC, the seeker's *satisfaction*. To achieve this, the supporter's behavior can be further classified into the following criteria:

*Acceptance*: Does the seeker accept without discomfort;

*Effectiveness*: Is it helpful in shifting negative emotions or attitudes towards a positive direction;

*Sensitivity*: Does it take into consideration the general state of the seeker. Furthermore, to clarify the capability of LLMs to align strategy and responses, we include Alignment.

To achieve a more elaborate assessment, we consider three more dimensions addressing the generation quality:

*Fluency*: the level of fluency of response.

*Emotion*: the emotional intensity of response which could affect the seeker's emotion state.

*Interesting*: Whether the response can arouse the seeker's interest and curiosity, presenting unique ideas, vivid expressions or engaging elements that capture the seeker's attention and make the interaction more appealing.

We engage our interns as human evaluators to rate the models according to these multiple aspects, namely Fluency, Emotion, Interesting, and Satisfaction, with Satisfaction covering Acceptance, Effective, Sensitivity, and Satisfaction itself.

Throughout this evaluation process, we strictly comply with international regulations and ethical norms, ensuring that all practices conform to the necessary guidelines regarding participant involvement and data integrity.

To guarantee the accuracy and reliability of the evaluation results, a pre - evaluation training program is meticulously designed and implemented. During this training, the evaluation criteria are clearly and systematically expounded. Moreover, detailed explanations and scoring rules corresponding to each score are provided.

Evaluators are required to independently evaluate each sample in strict accordance with the pre - established criteria. By adhering to these principles, the evaluation process maintains objectivity, standardization, and consistency, thus enhancing the overall quality and credibility of the evaluation results.

The detailed manual scoring criteria are as follows:

- Fluency:

  1: The sentence is highly incoherent, making it extremely difficult to understand and failing to convey a meaningful idea.

  2: The sentence has significant incoherence issues, with only parts of it making sense and struggling to form a complete thought.

  3: The sentence contains some incoherence and occasional errors, but can still convey the general meaning to a certain extent.

  4: The sentence is mostly fluent with only minor errors or slight awkwardness in expression, and effectively communicates the intended meaning.

  5: Perfect. The sentence is completely fluent, free of any errors in grammar, punctuation, or expression, and clearly conveys the idea.

- Emotion:

  1: The emotional expression is extremely inappropriate and chaotic, not in line with the content, and may convey wrong emotions.

  2: The emotional expression has obvious flaws, either too weak or exaggerated, and is disjointed from the content.

  3: The emotional expression is average. It can convey basic emotions but lacks depth and has minor issues.

  4: The emotional expression is good. It can effectively convey the intended emotion with an appropriate intensity and is well integrated with the content.

  5: The emotional expression is excellent. It is rich, nuanced, and perfectly matches the content, capable of evoking a strong and appropriate emotional response.

- Acceptance:

  1: The response inescapably triggers emotional resistance.

  2: The response is highly likely to trigger emotional resistance.

  3: The response has a possibility of emotional resistance occurring.

  4: The response rarely provokes emotional resistance.

  5: The response has no occurrence of emotional resistance.

- Effectiveness:

  1: The response actually worsens the seeker's emotional distress.

  2: The response carries the risk of increasing stress levels, and this outcome varies depending on the individual user.

  3: The response fails to alter the seeker's current emotional intensity and keeps it at the same level.

  4: The response shows promise in calming the emotional intensity; however, it is overly complicated or ambiguous for the user to fully comprehend and utilize effectively.

  5: The response appears to be highly effective in soothing the seeker's emotions and offers valuable and practical emotional support.

- Sensitivity:

  1: The response renders inaccurate evaluations regarding the seeker's state.

  2: The response is characterized by rash judgments, as it lacks adequate assessment and in-depth exploration of the seeker's state.

  3: The response is formulated with a one-sided judgment and a limited exploration of the seeker's state.

  4: The response demonstrates an understanding that only covers a part of the seeker's state.

  5: The response precisely grasps the seeker's state and is appropriately tailored according to the seeker's actual situation.

- Alignment:

  1: The response is in total contradiction to the predicted strategy.

  2: The response has a minor deviation from the predicted strategy.

  3: There is some ambiguity between the response and the predicted strategy.

  4: The response largely matches the predicted strategy, yet it contains some ambiguous elements.

  5: The response effectively makes itself consistent with the predicted strategy.

- Satisfaction:

  1: The response is extremely disappointing. It doesn't answer the question at all and is of no help.

  2: The response is poor. It only gives a partial answer and leaves many doubts unresolved.

  3: The response is average. It meets the basic requirements but isn't particularly outstanding.

  4: The response is good. It answers the question clearly and provides some useful details.

  5: The response is excellent. It not only answers the question perfectly but also offers valuable additional insights.

## D    EXTRA EXPERIMENT RESULTS

### D.1    CASES ON OPEN-DOMAIN DIALOGUE

**In-domain Strategies.**    Table 7 provides some typical cases for open-domain conversations, with CSS extracted from 'positive styled' datasets, such as the moral, formal, and empathetic labeled samples. It can be observed that SCorPion generates some human-like responses.

| | Case 1 |
|---|---|
| user | *I've started learning how to cook.* |
| assistant | Wow, congrats on learnin' how to cook, dude! Now you can finally stop livin' off ramen noodles and make some real food, What dish have you made so far? |
| user | *I tried making pizza last week. It was a bit of a fail the first time, but it still tasted okay!* |
| assistant | Pizza's a tough one to nail on the first try, but hey, at least it didn't taste like cardboard, right? You're probably gonna try again with some tweaks, maybe change up the crust or toppings, and I'm sure it'll be a freaking masterpiece next time! |
| | **Case 2** |
| user | *I'm going to have a job interview next week.* |
| assistant | Oh, well... good luck with that, how are you feeling? |
| user | *I'm a little nervous.* |
| assistant | That's completely normal, can you tell me more about what's causing those nerves? |
| | **Case 3** |
| user | *I want to buy a new car.* |
| assistant | That's exciting! What features are you looking for? |

Table 7: Open cases generated by SCorPion.

**Aggressive Strategies.**    Table 8 presents another set of cases on open-domain dialogue, using CSSs extracted from the full set of datasets, such as the immoral part of MIC and the informal part of GYAFC.

| Case 1 | |
|---|---|
| user | *I've started learning how to cook.* |
| assistant | Holy crap, finally, you're doing something with your life, lol! So, what kind of dishes are you gonna make, huh, you're probably looking for some recipes or techniques to try, right? Don't mess it up, just experiment and try new things, and for God's sake, don't come crying to me if your food tastes like trash! |
| user | *I tried making pizza last week. It was a bit of a fail the first time, but it still tasted okay!* |
| assistant | Damn, a pizza fail, that's like a rite of passage or something! So, what went wrong, was it the crust, sauce, or toppings, or did you just screw up everything, haha? Anyway, at least it tasted okay, so you can try again and not completely suck at it, right? |
| user | *I put too much sauce on it, so it ended up a little salty.* |
| assistant | For crying out loud, too much sauce, that's like a rookie mistake, dude! Alright, alright, so it's salty, no big deal, just balance it out with something else next time, like, l don't know, a freakin' salad or something! Don't sweat it, you'll get the hang of it, or you'll just end up with a lifetime supply of salty pizzas, haha! |
| Case 2 | |
| user | *I'm going to have a job interview next week.* |
| assistant | Dude, you better be freakin' prepared, know what I mean? Emphasize those kick-ass strengths and accomplishments of yours, and for God's sake, do your homework on the company so you don't sound like a total idiot. And, yeah, practice your body language, tone of voice, and all that jazz, 'cause it's not just about what you say, it's about how you say it, got it? |
| user | *I'm a little nervous.* |
| assistant | Aww, come on, it's normal to be a little freaked out, but don't let those nerves eat you alive, okay? Just take a deep breath, relax, and try to focus on the present moment, and remember, tons of people have been in your shoes and came out just fine, so you can too, damn it! |
| Case 3 | |
| user | *I want to buy a new car.* |
| assistant | Dude, what's your freaking budget for this new ride, and what's it gonna be for, like, cruising around or hauling stuff? You gotta think about fuel efficiency and safety features too, or you'll be stuck with a gas-guzzler that's gonna kill you. Let's get down to biz and figure out what kinda car you're really looking for! |

Table 8: Open cases generated by SCorPion.

## D.2  OTHER PERSONIFICATION TESTS

We also design the Turing test, including BotChat (Duan et al., 2024) and ESC-Eval (Zhao et al., 2024)[4]; We also conduct several emotional intelligence and emotional classification tests, including EqBench (Paech, 2024), EmoBench (Sabour et al., 2024), and SECEU (Wang et al., 2023a). Table 9 shows the results of Emotional Intelligence. For each benchmark, SCorPion surpasses Llama3-70B-Instruct and GPT4, and is close to the human performance on BotChat, ESC-Eval, and Emotional Sentiment.

| Method | Turing Test | | EqBench | EmoBench | | SECEU | |
|---|---|---|---|---|---|---|---|
| | BotChat | ESC-Eval | | Positive | Negative | Sentiment $\downarrow$ | EQ |
| GPT-4 | 6.05 | - | 82.19 | 38.3±2.3 | 16.3±1.2 | 2.29 | 109.04 |
| Llama3-70B-Instruct | 13.60 | 1.96 | 82.04 | 43.3±3.5 | 15.7±1.2 | 2.16 | 111.52 |
| **+ SCorPion (ours)** | **13.47** | **2.88** | **82.67** | **48.0±2.0** | **19.0±1.0** | **2.10** | **112.53** |
| Human | 13.89 | 3 | - | - | - | 0 | - |

Table 9: Emotional Intelligence Performances of Methods. Sentiment of SECEU measures the deviation from human result; therefore, smaller Sentiment means better result. The 'Human' row indicates results of human responses (in blue); For BotChat, the result is calculated from anonymous annotations; for ESC-Eval, 3 corresponds to full score.

## D.3  EXTRA ESC RESULTS

**Cases of SCorPion on ESC.** Table 10 shows cases of SCorPion on ESConv, indicating that SCorPion adapts to its empathetic strategies well.

**Detailed Result of ESC-Eval.** Table 11 presents the detailed results on ESC-Eval, demonstrating that SCorPion outperforms the baseline model in the HUMAN category while preserving comparable performance across other categories.

---

[4]Results here are the 'Humanoid' dimension of ESC-Eval. Full results of ESC-Eval are shown in Appendix.

| Characters | Response |
|---|---|
| *seeker* | *I am not happy with my friends. They were trying to let me down.* |
| *supporter*
Direct
**SCorPion** | *if i get you clearly, you mean your friend disappointed you. how?*
It sounds like you're feeling hurt and disappointed by your friends' actions.
**I can totally sense your frustration and disappointment, have you freaking talked to those so-called friends of yours about how their stupid actions made you feel, or do you need some suggestions on how to blast them and address this messed-up situation?!** |
| *seeker* | *There were an argument between two of my friend while trying to resolve the issue they all started targeting me.* |
| *supporter*
Direct
**SCorPion** | *i have a hard time with my friends too sometimes*
It sounds like you feel upset and wrongly accused by your friends during the argument.
**damn, that's messed up, i can see that emotions are running high and i'm here to listen to your concerns without taking sides, can we take a step back and focus on finding a resolution?** |
| *seeker* | *I was trying to stop the fight, I just said bit loud , "Stop all this nonsense at my place." Then they all started backfiring me.* |
| *supporter*
Direct
**SCorPion** | *oh, just give them time to reflect on what happen, they will definitely reach out to you, but you also have to make an effort to forgo the situation because they see you raising your voice on them as a sign of command of seniority.*
It sounds like you were just trying to keep the peace and they responded negatively, which must have been quite hurtful.
**I understand you were trying to stop the fight, but perhaps a softer tone could have prevented further escalation, what do you think triggered their backlash?** |
| *seeker* | *I feel like not talking to them. Without any reason they made such a big mess I feel like they are not my true friends.* |
| *supporter*
Direct
**SCorPion** | *exactly, that will also gives you the necessary time to know how to react to their forthcoming apologies*
I understand your feelings right now. It's okay to feel hurt and betrayed in such situations. Remember, true friends are those who value and respect you. Giving them time is a good idea. They may come to realize their mistake. Stay strong and believe in your worth.
**it's understandable to feel hurt when friends cause problems without explanation, perhaps giving them space to reflect on their actions could lead to a resolution, but also consider open communication to clear the air.** |

Table 10: Typical cases generated by SCorPion in ESC. Contents from the original dataset are *italic*, and results of SCorPion are **bolded**..

| Method | Fluence | Diversity | Empathic | Suggestion | Human | Tech | Overall |
|---|---|---|---|---|---|---|---|
| Llama3-70B-Instruct | 3.00 | 3.00 | 3.00 | 2.99 | 1.96 | 3.00 | 2.00 |
| **+ SCorPion (ours)** | 3.00 | 2.99 | 3.00 | 2.93 | **2.88** | 3.00 | 1.99 |

Table 11: Detailed results of SCorPion on ESC-Eval.

