# OpenReview forum: "Personified, Stylized and Controllable Conversation Based on Strategic GraphRAG"
_ICLR.cc/2026/Conference — ICLR 2026 Conference Withdrawn Submission_

### Official Review · Reviewer_eSUh · 2025-10-21

**Soundness:** 2
**Presentation:** 2
**Contribution:** 1
**Rating:** 2
**Confidence:** 4

**Summary:**

This paper proposes a conversation system called SCorPion for controllable, anthropomorphic, and stylized dialogue generation. The authors introduce GraphRAG to multi-turn dialogue generation, extracting and summarizing dialogue strategies (Community Summarized Strategies, CSS) from multi-source text corpora and generating responses using a top-k ranking mechanism during the query phase. The system allows for adjustment of the model's dialogue style through explicit control factors, such as empathy, formality, and moral inclination. Experiments were conducted on multiple dialogue and style tasks, including ESConv, DailyDialog, MIC, GYAFC, and Shakespeare. The results show that SCorPion outperforms various baselines (such as BOT, CSIM, and Self-Refine) in BLEU-2, ROUGE-L, and human evaluation.

**Strengths:**

1. The paper proposes using GraphRAG for dialogue policy retrieval and summarization, introducing knowledge graph structures into open-domain dialogue control, making generation structured and interpretable.
2. By introducing explicit control factors, the system can adjust the generated style (such as empathy, morality, and formality) based on external input.
3. The authors conducted experiments on multiple tasks and different model sizes (LLaMA-3-70B, GPT-3.5, etc.), and provided ablation analysis and sensitivity testing.

**Weaknesses:**

1. The authors report improvements in BLEU-2 and ROUGE-L, but these metrics are limited in their representativeness for conversational tasks and lack quantitative results on metrics closer to human perception (such as coherence, engagement, and consistency). While the human evaluation (Table 3) provides subjective metrics, it does not specify sample size and consistency (such as inter-annotator agreement), thus undermining these claims.
2. The paper claims to utilize GraphRAG at its core, but fails to clearly explain why the graph structure is necessary or how it compares to a standard RAG framework (such as the one used by the baseline BOT in the paper). The paper mentions using only top-level communities to generate CSS, raising questions about whether the method truly leverages GraphRAG's strengths in community discovery and graph traversal. While the current experimental results demonstrate that SCorPion outperforms BOT, this is likely due to the inherent effectiveness of the proposed CSS and its retrieval scoring mechanism (scored by the Mg model), and has little to do with GraphRAG's graph structure. The paper needs to provide more direct evidence demonstrating the necessity and superiority of the graph structure.
3. The authors present control curves in Figure 5 but do not provide any quantitative correlation analysis or automated evaluation metrics to demonstrate the causal consistency between the control factors and style changes. The results are primarily qualitative and lack statistical significance tests.
4. The paper does not clearly explain the fundamental differences between SCorPion and these methods in terms of theoretical mechanism or algorithmic complexity. The claim of GraphRAG's global retrieval capabilities alone cannot support its claimed conceptual advancement.

**Questions:**

1. Does the range of the control factor ccate require task-specific tuning? Will conflicts arise if different domains (e.g., empathy vs. formality) are controlled simultaneously?
2. In the human evaluation section, was inter-rater reliability (e.g., Cohen’s κ) calculated?
3. Please provide specific examples or statistical analysis to quantitatively correlate the control factor with variation in generated style (e.g., Pearson correlation or style classifier agreement).
4. Please describe the difference in inference latency and resource usage of SCorPion compared to BOT to support the claim of being lighter and faster.

---

### Official Review · Reviewer_nWtN · 2025-10-30

**Soundness:** 3
**Presentation:** 3
**Contribution:** 3
**Rating:** 6
**Confidence:** 3

**Summary:**

This paper introduces SCorPion, a dialogue AI agent based on GraphRAG that generates personalized, stylized, and controllable responses. The system aims to overcome the limitations of fixed-prompt methods by extracting dialogue strategies from a variety of text corpora, allowing the dialogue style to be adjusted according to different needs. The paper’s goal is to enhance open-domain conversations, emotional support interactions, and style adaptation through control based on dynamic strategies. The main contribution is the dynamic retrieval of strategies from a knowledge graph to guide large language models (LLMs) in generating more adaptive and personalized dialogues.

**Strengths:**

1. The innovative integration of GraphRAG into the dialogue system generates response strategies based on a knowledge graph, enhancing the flexibility and depth of the dialogue model.

2. SCorPion allows for explicit control over response styles (such as empathy, formality, etc.), enabling it to adapt to different conversational contexts.

3. The paper provides detailed experiments demonstrating the superiority of SCorPion in tasks such as emotional support and style adaptation, proving that this approach outperforms other baseline methods in emotional intelligence and stylized responses.

**Weaknesses:**

1. Although the proposed method is effective, the global summarization process of GraphRAG may incur high computational costs, which could become a limitation in practical applications. Additionally, the paper primarily focuses on automated testing and controlled experiments, lacking exploration of user interactions in more realistic and unpredictable scenarios. There is insufficient discussion on practical application issues, such as computational overhead, real-time conversation efficiency, and user acceptance.

2. The paper introduces a ranking mechanism to control output styles. Further clarification is needed on how to fine-tune these control factors and their practical application across various scenarios.

**Questions:**

1. Could you provide more details on the trade-off between computational efficiency and flexibility in the GraphRAG model? Specifically, how does the top-k ranking mechanism balance performance and resource consumption in real-time applications?

2. What measures have you taken to reduce bias in the dialogue strategies retrieved by GraphRAG? Is it possible to introduce more refined filtering mechanisms to enhance fairness in sensitive topics?

---

### Official Review · Reviewer_cJWM · 2025-11-03

**Soundness:** 2
**Presentation:** 2
**Contribution:** 2
**Rating:** 4
**Confidence:** 2

**Summary:**

The authors propose SCorPion, a conversational model that uses GraphRAG to generate controllable, personalized, and stylized responses by dynamically retrieving conversational strategies from multiple datasets (e.g., DailyDialog, ESConv, MIC, GYAFC). The system modifies GraphRAG to index in the format {Condition (when to act), Skill (how to act), Demonstration (examples)} and implements a top-k ranking mechanism for computational efficiency. A key innovation is explicit controllability through control factors that re-weight the Condition/Skill/Demonstration scores to adjust response styles (empathy, formality, morality). The authors compare their approach against prompting baselines (Direct, Self-Refine, CoT, BOT) across open-domain dialogue, emotional support, and style adaptation tasks. SCorPion shows better performance on both human evaluations (Satisfaction: 3.57 vs 3.00, Effectiveness: 3.50 vs 3.00, Sensitivity: 3.90 vs 3.20) and automatic metrics (BLEU-2, ROUGE-L).

**Strengths:**

-  Repurposing GraphRAG from knowledge-intensive QA to conversational strategy retrieval is novel through explicit reweighting factors, though it primarily combines existing ideas rather than introducing fundamental innovations.
- The paper covers many benchmarks tasks (open-domain dialogue, emotional support, style adaptation, safety, emotional intelligence) showing consistent improvements.
- The paper is mostly well-written but presentation can be improved since it refers a lot to the appendix reliance.

**Weaknesses:**

- The paper's is incremental, with minor improvements (BLEU-2: 4.32 vs 3.44 = 0.88 gain, ~20% relative), it just show better retrieval mechanism, which per-se is not novel. Moreover, the approach is dataset-dependent, requiring domain-specific indexing with manual intervention, which limits generalizability and suggests the method may not scale beyond selected datasets to become a real open-domain model.

- The paper does not compare against state-of-the-art frontier models (GPT-4, Claude-3, Gemini-Pro) that may achieve superior performance without any retrieval overhead. This is important because: (1) strong LLMs has conversational strategies through pre-training and RLHF, (2) and directly using one LLM would be significantly faster than two-stage process (GraphRAG retrieval + LLaMA3-70B generation),

**Questions:**

What is the end-to-end latency comparison between SCorPion (GraphRAG retrieval + generation) versus direct calling any frontier model API inference? Can you provide a cost-benefit analysis showing retrieval overhead (computational cost, latency) versus performance gains?

---

### Note · Authors · 2025-11-16

I have read and agree with the venue's withdrawal policy on behalf of myself and my co-authors.